# Learning Hyperbolic Representations of Topological Features

**Panagiotis Kyriakis**
University of Southern California
Los Angeles, USA
pkyriaki@usc.edu

**Iordanis Fostiropoulos**
University of Southern California
Los Angeles, USA
fostirop@usc.edu

**Paul Bogdan**
University of Southern California
Los Angeles, USA
pbogdan@usc.edu

## Abstract

Learning task-specific representations of persistence diagrams is an important problem in topological data analysis and machine learning. However, current methods are restricted in terms of their expressivity as they are focused on Euclidean representations. Persistence diagrams often contain features of infinite persistence (i.e., *essential features*) and Euclidean spaces shrink their importance relative to non-essential features because they cannot assign infinite distance to finite points. To deal with this issue, we propose a method to learn representations of persistence diagrams on hyperbolic spaces, more specifically on the Poincare ball. By representing features of infinite persistence infinitesimally close to the boundary of the ball, their distance to non-essential features approaches infinity, thereby their relative importance is preserved. This is achieved without utilizing extremely high values for the learnable parameters, thus, the representation can be fed into downstream optimization methods and trained efficiently in an end-to-end fashion. We present experimental results on graph and image classification tasks and show that the performance of our method is on par with or exceeds the performance of other state of the art methods.

## 1 Introduction

*Persistent homology* is a topological data analysis tool which tracks how topological features (e.g. connected components, cycles, cavities) appear and disappear as we analyze the data at different scales or in nested sequences of subspaces (1; 2). A nested sequence of subspaces is known as a *filtration*. As an informal example of a filtration consider an image of variable brightness. As the brightness is increased, certain features (edges, texture) may become less or more prevalent. The *birth* of a topological feature refers to the "time" (i.e., the brightness value) when it appears in the filtration and the *death* refers to the "time" when it disappears. The lifespan of the feature is called *persistence*. Persistent homology summarizes these topological characteristics in a form of multiset called *persistence diagram*, which is a highly robust and versatile descriptor of the data. Persistence diagrams enjoy the *stability* property, which ensures that the diagrams of two similar objects are similar (3). Additionally, under some assumptions, one can approximately reconstruct the input space from a diagram (which is known as solving the inverse problem) (4). However, despite their strengths, the space of persistence diagrams lacks structure as basic operations, such as addition and scalar multiplication, are not well defined. The only imposed structure is induced by the Bottleneck and Wasserstein metrics, which are notoriously hard to compute, thereby preventing us from leveraging them for machine learning tasks.

**Related Work**. To address these issues, several vectorization methods have been proposed. Some of the earliest approaches are based on *kernels*, i.e., generalized products that turn persistence diagrams into elements of a Hilbert space. Kusano et al. (5) propose a persistence weighted Gaussian kernel which allows them to explicitly control the effect of persistence. Alternatively, Carrière et al. (6) leverage the sliced Wasserstein distance to define a kernel that mimics the distance between diagrams. The approaches by Bubenik (7) based on persistent landscapes, by Reininghaus et al. (8) based on scale space theory and by Le et al. (9) based on the Fisher information metric are along the same line of work. The major drawback in utilizing kernel methods is that they suffer from scalability issues as the training scales poorly with the number of samples.

In another line of work, researchers have constructed *finite-dimensional embeddings*, i.e., transformations turning persistence diagrams into vectors in a Euclidean space. Adams et al. (10) map the diagrams to persistence images and discretize them to obtain the embedding vector. Carrière et al. (11) develop a stable vectorization method by computing pairwise distances between points in the persistence diagram. An approach based on interpreting the points in the diagram as roots of a complex polynomial is presented by Di Fabio (12). Adcock et al. (13) identify an algebra of polynomials on the diagram space that can be used as coordinates and the approach is extended by Kališnik in (14) to tropical functions which guarantee stability. The common drawback of these embeddings is that the representation is pre-defined, i.e., there exist no learnable parameters, therefore, it is agnostic to the specific learning task. This is clearly sub-optimal as the eminent success of deep learning has demonstrated that it is preferable to learn the representation.

The more recent approaches aim at learning the representation of the persistence diagram in an end-to-end fashion. Hofer et al. (15) present the first input layer based on a parameterized family of Gaussian-like functionals, with the mean and variance learned during training. They extend their method in (16) allowing for a broader class of parameterized function families to be considered. It is quite common to have topological features of infinite persistence (1), i.e., features that never die. Such features are called *essential* and in practice are usually assigned a death time equal to the maximum filtration value. This may restrict their expressivity because it shrinks their importance relative to non-essential features. While we may be able to increase the scale sufficiently high and end up having only one trivial essential feature (i.e., the 0-th order persistent homology group that becomes a single connected component at a scale that is sufficiently large), the resulting persistence diagrams may not be the ones that best summarize the data in terms of performance on the underlying learning task. This is evident in the work by Hofer et al. (15) where the authors showed that essential features offer discriminative power. The work by Carrière et al. (17), which introduces a network input layer the encompasses several vectorization methods, emphasizes the importance of essential features and is the first one to introduce a deep learning method incorporating extended persistence as a way to deal with them.

In this paper, we approach the issue of essential features from the geometric viewpoint. We are motivated by the recent success of hyperbolic geometry and the interest in extending machine learning models to hyperbolic spaces or general manifolds. We refer the reader to the review paper by Bronstein et al. (18) for an overview of geometric deep learning. Here, we review the most relevant and pivotal contributions in the field. Nickel et al. (19; 20) propose Poincaré and Lorentz embeddings for learning hierarchical representations of symbolic data and show that the representational capacity and generalization ability outperform Euclidean embeddings. Sala et al. (21) propose low-dimensional hyperbolic embeddings of hierarchical data and show competitive performance on WorldNet. Ganea et al. (22) generalize neural networks to the hyperbolic space and show that hyperbolic sentence embeddings outperform their Euclidean counterparts on a range of tasks. Gulcherhe et al. (23) introduce hyperbolic attention networks which show improvements in terms of generalization on machine translation and graph learning while keeping a compact representation. In the context of graph representation learning, hyperbolic graph neural networks (24) and hyperbolic graph convolutional neural networks (25) have been developed and shown to lead to improvements on various benchmarks. However, despite this success of geometric deep learning, little work has been done in applying these methods to topological features, such as persistence diagrams.

The **main contribution** of this paper is to bridge the gap between topological data analysis and hyperbolic representation learning. We introduce a method to represent persistence diagrams on a hyperbolic space, more specifically on the Poincare ball. We define a learnable parameterization of the Poincare ball and leverage the vectorial structure of the tangent space to combine (in a manifold-preserving manner) the representations of individual points of the persistence diagram. Our method learns better task-specific representations than the state of the art because it does not shrink the relative importance of essential features. In fact, by allowing the representations of essential features to get infinitesimally close to the boundary of the Poincare ball, their distance to the representations of non-essential features approaches infinity, therefore preserving their relative importance. To the best of our knowledge, this is the first approach for learning representations of persistence diagrams in non-Euclidean spaces.

## 2  BACKGROUND

In this section, we provide a brief overview of persistent homology leading up to the definition of persistence diagrams. We refer the interested reader to the papers by Edelsbrunner et al. (1; 2) for a detailed overview of persistent homology. An overview of homology can be found in the Appendix.

**Persistent Homology**. Let $K$ be a simplicial complex. A *filtration* of $K$ is a nested sequence of subcomplexes that starts with the empty complex and ends with $K$,

$$\emptyset = K_0 \subseteq K_1 \subseteq \ldots \subseteq K_d = K. \tag{1}$$

A typical way to construct a filtration is to consider sublevel sets of a real valued function, $f : K \to \mathbb{R}$. Let $a_1 < \cdots < a_d$ be a sorted sequence of the values of $f(K)$. Then, we obtain a filtration by setting

$$K_0 = \emptyset \quad \text{and} \quad K_i = f^{-1}((-\infty, a_i]) \text{ for } 1 \leq i \leq d. \tag{2}$$

We can apply simplicial homology to each of the subcomplexes of the filtration. When $0 \leq i \leq j \leq d$, the inclusion $K_i \subseteq K_j$ induces a homomorphism

$$f_n^{i,j} : H_n(K_i) \to H_n(K_j) \tag{3}$$

on the simplicial homology groups for each homology dimension $n$. We call the image of $f_n^{i,j}$ a *n-th persistent homology group* and it consists of homology classes born before $i$ that are still alive at $j$. A homology class $\alpha$ is *born* at $K_i$ if it is not in the image of the map induced by the inclusion $K_{i-1} \subseteq K_i$. Furthermore, if $\alpha$ is born at $K_i$, it dies entering $K_j$ if the image of the map induced by $K_{i-1} \subseteq K_{j-1}$ does not contain the image of $\alpha$ but the image of the map induced by $K_{i-1} \subseteq K_j$ does. The *persistence* of the homology class $\alpha$ is $j - i$. Since classes may be born at the same $i$ and die at the same $j$, we can use inclusion-exclusion to determine the multiplicity of each $(i, j)$,

$$\mu_n^{i,j} = \beta_n^{i,j-1} - \beta_n^{i-1,j-1} - \beta_n^{i,j} + \beta_n^{i-1,j}, \tag{4}$$

where the *n-th persistent Betti numbers* $\beta_n^{i,j}$ are the ranks of the images of the $n$-th persistent homology group, i.e., $\beta_n^{i,j} = rank(im(f_n^{i,j}))$, and capture the number of $n$-dimensional topological features that persist from $i$ to $j$. By setting $\mu_n^{i,\infty} = \beta_n^{i,d} - \beta_n^{i-1,d}$ we can account for features that still persist at the end of the filtration ($j = d$), which are known as *essential features*.

**Persistence Diagrams**. Persistence diagrams are multisets supported by the upper diagonal part of the real plane and capture the birth/death of topological features (i.e., homology classes) across the filtration.

**Definition 2.1** (Persistence Diagram). Let $\Delta = \{x \in \mathbb{R}_\Delta : mult(x) = \infty\}$ be the multiset of the diagonal $\mathbb{R}_\Delta = \{(x_1, x_2) \in \mathbb{R}^2 : x_1 = x_2\}$, where $mult(\cdot)$ denotes the multiplicity function and let $\mathbb{R}_*^2 = \{(x_1, x_2) \in \mathbb{R} \cup (\mathbb{R} \cup \infty) : x_2 > x_1\}$. Also, let $n$ be a homology dimension and consider the sublevel set filtration induced by a function $f : K \to \mathbb{R}$ over the complex $K$. Then, a *persistence diagram*, $\mathcal{D}_n(f)$, is a multiset of the form $\mathcal{D}_n(f) = \{x : x \in \mathbb{R}_*^2\} \cup \Delta$ constructed by inserting each point $(a_i, a_j)$ for $i < j$ with multiplicity $\mu_n^{i,j}$ (or $\mu_n^{i,\infty}$ if it is an essential feature). We denote the space of all persistence diagrams with $\mathbb{D}$.

**Definition 2.2** (Wasserstein distance and stability). Let $\mathcal{D}_n(f), \mathcal{E}_n(g)$ be two persistence diagrams generated by the filtration induced by the functions $f, g : K \to \mathbb{R}$, respectively. We define the *Wasserstein distance*

$$w_p^q(\mathcal{D}_n(f), \mathcal{E}_g(g)) = \inf_{\eta} \big( \sum_{x \in \mathcal{D}} \|x - \eta(x)\|_q^p \big)^{1/p}, \tag{5}$$

where $p, q \in \mathbb{N}$ and the infimum is taken over all bijections $\eta : \mathcal{D}_n(f) \to \mathcal{E}_n(g)$. The special case $p = \infty$ is known as *Bottleneck distance*. The persistence diagrams are stable with respect to the Wasserstein distance if and only if $w_p^q(\mathcal{D}_n(f), \mathcal{E}_g(g)) \leq \|f - g\|_\infty$.

Note that a bijection $\eta$ between persistence diagrams is guaranteed to exist because their cardinalities are equal, considering that, as per Def. 2.1, the points on the diagonal are added with infinite multiplicity. The strength of persistent homology stems from the above stability definition, which essentially states that the map taking a sublevel function to the persistence diagram is Lipschitz continuous. This implies that if two objects are similar then their persistence diagrams are close.

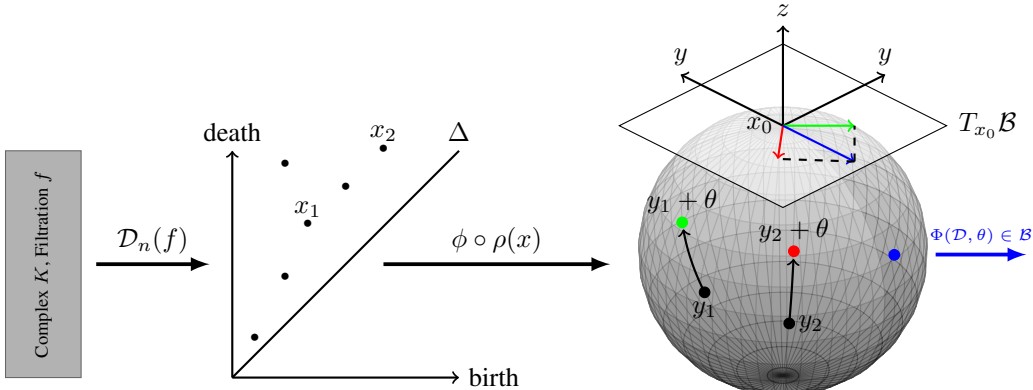

Figure 1: **Illustration of our method**: Initially, the points are transferred via the auxiliary transformation $\rho$ and the parameterization $\phi$ to the Poincare ball $\mathcal{B}$, where learnable parameters $\theta$ are added. Then, the logarithmic map is used for transforming the points to the tangent space $T_{x_0}\mathcal{B}$. Finally, the resulting vectors are added and transformed back to the manifold via the exponential map. Note that the persistence diagram is mapped to a single point on the Poincare ball (i.e., $\Phi(\mathcal{D}, \theta) \in \mathcal{B}$).

## 3 PERSISTENT POINCARE REPRESENTATIONS

In this section, we introduce our method (Fig. 1) for learning representations of persistence diagrams on the Poincare ball. We refer the reader to the Appendix for some fundamental concepts of differential geometry.

The Poincare ball is an $m$-dimensional manifold $(\mathcal{B}, g_x^{\mathcal{B}})$, where $\mathcal{B} = \{x \in \mathbb{R}^m : \|x\| < 1\}$ is the open unit ball. The space in which the ball is embedded is called *ambient space* and is assumed to be equal to $\mathbb{R}^m$. The Poincare ball is conformal (i.e., angle-preserving) to the Euclidean space but it does not preserve distances. The metric tensor and distance function are as follows

$$g_x^{\mathcal{B}} = \lambda_x^2 g^E \quad \lambda_x = \frac{2}{1 - \|x\|^2} \quad d_{\mathcal{B}}(x, y) = \arccos\left(1 + 2\frac{\|x - y\|^2}{(1 - \|x\|^2)(1 - \|y\|)^2}\right), \quad (6)$$

where $g^E = \mathbb{I}_m$ is the Euclidean metric tensor. Eq. 6 highlights the benefit of using the Poincare ball for representing persistence diagrams. Contrary to Euclidean spaces, distances in the Poincare ball can approach infinity for finite points. This space is ideal for representing essential features appearing in persistence diagrams without squashing their importance relative to non-essential features. Informally, this is achieved by allowing the representations of the former ones to get infinitesimally close to the boundary, thereby their distances to the later ones approach infinity. Fig. 2 provides an illustration.

We gradually construct our representation through a composition of 3 individual transformations. The **first step** is to transfer the points to the ambient space (i.e., $\mathbb{R}^m$) of the Poincare ball. Let $\mathcal{D}^1$ be a persistence diagram. We introduce the following *auxiliary transformation*

$$\rho : \mathbb{R}_*^2 \to \mathbb{R}^m. \quad (7)$$

This auxiliary transformation is essentially a high-dimensional embedding and may contain learnable parameters. Nonetheless, our main focus is to learn a hyperbolic representation and, therefore, we assume that $\rho$ is not learnable. Later in this section, we analyze conditions on $\rho$ to guarantee the stability and expressiveness of the hyperbolic representation.

The **second step** is to transform the embedded points from the ambient space to the Poincare ball. When referring to points on a manifold, it is important to define a coordinate system. A homeomorphism $\psi : \mathcal{B} \to \mathbb{R}^m$ is called *coordinate chart* and gives the *local coordinates* on the manifold. The inverse map $\phi : \mathbb{R}^m \to \mathcal{B}$, is called a *parameterization* of $\mathcal{B}$ and gives the *ambient coordinates*. The main idea is to inject learnable parameters into this parameterization. The injected parameters could be any form of differentiable functional that preserves the homomorphic property. Differentiability is needed such that our representation can be fed to downstream optimization

---

[1]The sublevel set function $f$ and the homology dimension $n$ are omitted.

methods. In our construction, we utilize a variant of the generalized spherical coordinates. Let $\theta \in \mathbb{R}^m$ be a vector of $m$ parameters. We define the *learnable parameterization* $\phi : \mathbb{R}^m \times \mathbb{R}^m \to \mathcal{B}$ as follows

$$y_1 = 1 + \frac{2}{\pi} \arctan \theta_1 r_1 \text{ and } y_i = \theta_i + \arccos \frac{x_{i-1}}{r_{i-1}}, \text{ for } i = 2, 3, ..m, \tag{8}$$

where $r_i^2 = x_m^2 + \cdots + x_{i+1}^2 + x_i^2 + \epsilon$. The small positive constant $\epsilon$ is added to ensure that the denominator in Eq. 8 is not zero. Intuitively, Eq. 8 corresponds to scaling the radius of the point by a factor $\theta_1$ and rotating it by $\theta_i$ radians across the angular axes. The scaling and rotation parameters are learned during training. Note that the form of $y_1$ ensures that representation belongs in the unit ball for all values of $\theta_1$. The coordinate chart is not explicitly used in our representation; it is provided in the Appendix for the sake of completeness.

The **third step** is to combine the representations of each individual point of the persistence diagram into a single point in the hyperbolic space. Typically, in Euclidean spaces, this is done by concatenating or adding the corresponding representations. However, in non-Euclidean spaces such operations are not manifold-preserving. Therefore, we transform the points from the manifold to the tangent space, combine the vectors via standard vectorial addition and transform the resulting vector back to the manifold. This approach is based on the *exponential* and *logarithmic* maps

$$\exp_x : T_x\mathcal{B} \to \mathcal{B} \quad \text{and} \quad \log_x : \mathcal{B} \to T_x\mathcal{B}. \tag{9}$$

The exponential map allows us to transform a vector from the tangent space to the manifold and its inverse (i.e., the logarithmic map) from the manifold to the tangent space. For a general manifold, it is hard to find these maps as we need to solve for the minimal geodesic curve (see Appendix for more details). Fortunately, for the Poincare ball case, they have analytical expressions, given as follows

$$\exp_x(v) = x \oplus \left( \tanh\left(\frac{\lambda_x \|v\|}{2}\right) \frac{v}{\|v\|} \right), \log_x(y) = \frac{2}{\lambda_x} \tanh^{-1} \|-x \oplus y\| \frac{-x \oplus y}{\|-x \oplus y\|}, \tag{10}$$

where $\oplus$ denotes the Möbius addition, which is a manifold-preserving operator (i.e., for any $x, y \in \mathcal{B} \implies x \oplus y \in \mathcal{B}$). The analytical expression is given in the Appendix. The transformations given by these maps are norm-preserving, i.e., for example, the geodesic distance from $x$ to the transformed point $\exp_x(v)$ coincides with the metric norm $\|v\|_g$ induced by the metric tensor $g_x^{\mathcal{B}}$. This is an important property as we need the distance between points (and therefore the relative importance of topological features) to be preserved when transforming to and from the tangent space. We now combine the aforementioned transformations and define the Poincare hyperbolic representation followed by its stability theorem.

**Definition 3.1** (Poincare Representation). Let $\mathcal{D} \in \mathbb{D}$ be the persistence diagram to be represented in an $m$-dimensional Poincare ball $(\mathcal{B}, g_x^{\mathcal{B}})$ embedded in $\mathbb{R}^m$ and $x_0 \in \mathcal{B}$ be a given point. The representation of $\mathcal{D}$ on the manifold $\mathcal{B}$ is defined as follows

$$\Phi : \mathbb{D} \times \mathbb{R}^m \to \mathcal{B}, \quad \Phi(\mathcal{D}, \theta) = \exp_{x_0}\left( \sum_{x \in \mathcal{D}} \log_{x_0}\left( \phi(\rho(x)) \right) \right), \tag{11}$$

where the exponential and logarithmic maps are given by Eq. 10 and the learnable parameterization and the auxiliary transformation by Eq. 8 and Eq. 7, respectively.

**Theorem 1** (Stability of Hyperbolic Representation). Let $\mathcal{D}, \mathcal{E}$ be two persistence diagrams and consider an auxiliary transformation $\rho : \mathbb{R}_*^2 \to \mathbb{R}^m$ that is

- Lipschitz continuous w.r.t the induced metric norm $\|\cdot\|_g$,

- $\rho(x) = 0$ for all $x \in \mathbb{R}_\Delta$.

Additionally, assume that $x_0 = 0$. Then, the hyperbolic representation given by Eq. 11 is stable w.r.t the Wasserstein distance when $p = 1$, i.e., there exists constant $K > 0$ such that

$$d_\mathcal{B}(\Phi(\mathcal{D}, \theta), \Phi(\mathcal{E}, \theta)) \leq K w_1^g(\mathcal{D}, \mathcal{E}) \tag{12}$$

where $d_\mathcal{B}$ is the geodesic distance and $w_1^g$ is the Wasserstein metric with the $q$-norm replaced by the induced norm $\|\cdot\|_g$ (i.e., the norm induced by the metric tensor $g$, see Appendix A.2).

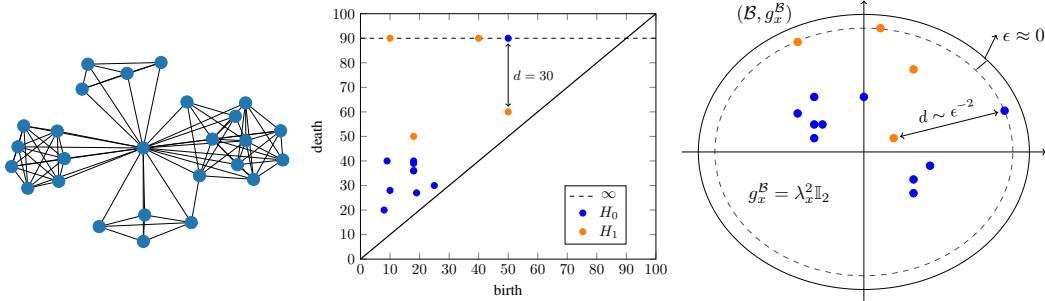

Figure 2: *Left*: Example graph from the IMDB-BINARY dataset. *Middle*: Persistence diagrams extracted using the Vietoris-Rips filtration. The dashed line denotes features of infinite persistence, which are represented by points of maximal death value equal to 90 (i.e., by points of finite persistence). *Right*: Equivalent representation on the 2-dimensional Poincare ball. Features of infinite persistence are mapped infinitesimally close to the boundary. Therefore, their distance to finite persistence features approaches infinity ($d \sim \epsilon^{-2}$).

The proof of Theorem 1 (given in the Appendix) results from a general stability theorem (3) and is on par with similar results for other vectorizations (10) or representations (15) of persistence diagrams. One subtle difference is that Theorem 1 uses the induced norm rather than the $q$-norm appearing in the Wasserstein distance. However, since the induced norm implicitly depends on the chosen point $x_0$, which, per requirements of Theorem 1, is assumed to be equal to the origin, there is no substantial difference. The fact that we require the auxiliary transformation $\rho$ to be zero on the diagonal is important to theoretically guarantee stability. Intuitively, this can be understood by recalling (Def. 2.1) that all (infinite) points on the diagonal are included in the persistence diagram. By mapping the diagonal to zero and taking $x_0 = 0$, we ensure that the summation in Eq. 11 collapses to zero when summing over the diagonal.

Finally, we note that the assumptions of Theorem 1 are not restrictive. In fact, we can easily find Lipschitz continuous transformations that are zero on the diagonal $\mathbb{R}_\Delta$, such as the exponential and rational transformations proposed by Hofer et al. (15). Additionally, we note that the assumptions of Theorem 1 do not prohibit us from choosing an "under-powered" or degenerate $\rho$. For example, $\rho = 0$ satisfies the assumptions and therefore leads to a stable representation. However, such representation is obviously not useful for learning tasks. An implicit requirement, that guarantees not only the stability but the expressiveness of the results representation, is that $\rho$ does not result in any information loss. This requirement is satisfied by picking a $\rho$ that it is injective, which, given that it is a higher dimensional embedding, is a condition easy to satisfy. In practice, we use a variant of the exponential transformation by Hofer et al. (15). The exact expression is given in the Appendix.

## 4 EXPERIMENTS

We present experiments on diverse datasets focusing on persistence diagrams extracted from graphs and grey-scale images. The learning task is classification. Our representation acts as an input to a neural network and the parameters are learned end-to-end via standard gradient methods. The architecture as well as other training details are discussed in the Appendix. The code to reproduce our experiments is publicly available at `https://github.com/pkyriakis/permanifold/`.

**Ablation Study**: To highlight to what extent our results are driven by the hyperbolic embedding, we perform an ablation study. In more detail, we consider three variants of our method:

1. Persistent Poincare (P-Poinc): This is the original method as presented in Sec. 3,

2. Persistent Hybrid (P-Hybrid): Same as P-Poinc with the Poincare ball replaced by the Euclidean space. This implies that the exponential and logarithmic maps (Eq. 10) reduce to the identity maps, i.e., $\exp_x(v) = x + v \log_x(y) = y - x$. The learnable parameterization is as in Eq. 8.

3. Persistent Euclidean (P-Eucl): Same as P-Hybrid with Eq. 8 replaced with simple addition of the learnable parameters, i.e., $y = x + \theta$.

**Baseline - Essential Features Separation**: To highlight the benefit of a unified Poincare representation, we design a natural baseline that treats essential and non-essential features separately. In more detail, for each point $(b, d) \in \mathcal{D}$, we calculate its persistence $d - b$ and then compute the histogram of the resulting persistence values. For essential features, we compute the histogram of their birth times. Then, we concatenate those histograms and feed them as input to the neural network (architecture described in the Appendix). We consider the case where the essential features are included (*baseline w/ essential*) and the case where they are discarded (*baseline w/o essential*).

**Manifold Dimension and Projection Bases**: Since our method essentially represents each persistence diagram on a $m-$dimensional Poincare ball, it may introduce substantial information compression when the points in the diagrams are not of the same order as $m$. A trivial approach to counteract this issue is to use a high value for $m$. However, we experimentally observed that a high manifold dimension does not give the optimal classification performance and it adds a computational overhead in the construction of the computation graph. Empirically, the best approach is to keep $m$ at moderate values (in the range $m = 3$ to $m = 12$), replicate the representation $K$ times and concatenate the outputs. Each replica is called a *projection base* and for their number we explored values dependant on the number of points in the persistence diagram. Persistence diagrams obtained from images tend to have substantially fewer points than diagrams obtained from graphs. Therefore, for images, we explored moderate values for $K$, i.e., $5 - 10$, whereas for graphs we increased $K$ in the range $200 - 500$. Essentially, we treat both $m$ and $K$ as hyper-parameters, explore their space following the aforementioned empirical rules and pick the optimal via the validation dataset. As we increase $m$, it is usually prudent to decrease $K$ to maintain similar model capacity.

### 4.1 GRAPH CLASSIFICATION

In this experiment, we consider the problem of graph classification. We evaluate our approach using social graphs from (26). The *REDDIT-BINARY* dataset contains 1000 samples and the graphs correspond to online discussion threads. The task is to identify to which community a given graph belongs. The *REDDIT-5K* and the *REDDIT-12K* are a larger variant of the former dataset and contain 5K and ~12K graphs from 5 and 11 subreddits, respectively. The task is to predict to which subreddit a given discussion graph belongs. The IMDB-BINARY contains 1000 ego-networks of actors that have appeared together in any movie and the task is to identify the genre (*action* or *romance*) to which an ego-graph belongs. Finally, the IMDB-MULTI contains 1500 ego-networks belonging to 3 genres (*action*, *romance*, *sci-fi*). We train our model using 10-fold cross-validation with a 80/20 split.

Graphs are special cases of simplicial complexes, therefore, defining filtrations is straightforward. We use two methods: The first one captures global topological properties and is based on shortest paths. In this case, the graph $\mathcal{G} = (V, E)$ is lifted to a metric space $(V, d)$ using the shortest path distance $d : V \times V \to \mathbb{R}_{\geq 0}$ between two vertices, which can be easily proved to be a valid metric. Then, we define the Vietoris-Rips complex of $\mathcal{G}$ as the filtered complex $VR_s(\mathcal{G})$ that contains a subset of $V$ as a simplex if all pairwise distances in that subset are less than or equal to $s$, or, formally,

$$VR_s(\mathcal{G}) = \{(v_0, v_1, \cdots, v_m) : d(v_i, v_j) \leq s, \forall i, j\}. \tag{13}$$

This approach essentially interprets the vertices of the graph as a point cloud in a metric space, with the distance between points given by the shortest path between the corresponding vertices. In the case of unweighted graph, we assign unit weight across edges. In Fig. 2 we show a sample persistence diagram extracted with the Vietoris-Rips filtration and the corresponding representation of the features on the Poincare ball. The second method captures local topological properties and is based on vertex degree. Given a graph $\mathcal{G} = (V, E)$, a simplicial complex can be defined as the union of the vertex and edge sets, i.e., $K = K_0 \cup K_1$, where $K_0 = \{v : v \in V\}$ and $K_1 = \{(u, v) : (u, v) \in E\}$. The sublevel function is defined as follows: $f(v) = deg(v)$ for $v \in K_0$ and $f(u, v) = \max\{f(u), f(v)\}$ for $(u, v) \in K_1$, where $deg(v)$ is the vertex degree $v$. Then, the filtration is given by Eq. 2.

We compare our method against several state of the art methods for graph classification. In particular, we compare against : (1) the Weisfeiler-Lehman (WL) graph kernel (27), (2) the Deep Graph Kernel (DGK) (26), (3) the Patchy-SAN (PSCN) (28), the Weighted Persistence Image Kernel (WKPI) (29), (5) the Anonymous Walk Embeddings (AWE) (30), (6) the Graph Isomorphism Network with Graph Filitraton Learning (GIN-GFL) (31). Additionally, we compare against the PersLay input layer by Carrière et al. (17) which utilizes extended persistence as an alternative way to deal with essential features and the Learnable Representation based on Gaussian-like structure elements (GLR) presented by Hofer et al. (15).

Table 1: Classification accuracy (mean±std or min-max range, if available).

| | IMDB-M | IMDB-B | REDDIT-B | REDDIT-5K | REDDIT-12K |
|---|---|---|---|---|---|
| *baseline w/o ess.* | $38.43_{\pm0.98}$ | $65.78_{\pm1.25}$ | $67.33_{\pm1.55}$ | $39.30_{\pm1.12}$ | $28.35_{\pm1.11}$ |
| *baseline w/ ess.* | $38.78_{\pm0.85}$ | $66.56_{\pm1.13}$ | $69.33_{\pm0.75}$ | $38.45_{\pm0.98}$ | $30.43_{\pm0.75}$ |
| WL | $49.33_{\pm4.75}$ | $73.40_{\pm4.63}$ | $81.10_{\pm1.90}$ | $49.44_{\pm2.36}$ | $38.18_{\pm1.31}$ |
| DGK | $44.5_{\pm0.52}$ | $66.96_{\pm0.56}$ | $78.04_{\pm0.39}$ | $41.27_{\pm0.18}$ | $32.22_{\pm0.10}$ |
| PSCN | $45.23_{\pm2.84}$ | $71.00_{\pm2.29}$ | $86.30_{\pm1.58}$ | $49.10_{\pm0.70}$ | $41.32_{\pm0.32}$ |
| WKPI | $49.5_{\pm0.40}$ | $75.10_{\pm1.10}$ | *n/a* | $59.50_{\pm0.60}$ | $48.40_{\pm0.50}$ |
| AWE | $51.54_{\pm3.61}$ | $74.45_{\pm5.83}$ | $87.89_{\pm2.53}$ | $54.74_{\pm1.91}$ | $39.20_{\pm2.0}$ |
| GIN-GFL | $49.70_{\pm2.9}$ | $74.50_{\pm4.60}$ | $90.30_{\pm2.60}$ | $55.70_{\pm2.90}$ | *n/a* |
| PersLay | 48.8-52.2 | 71.2-72.6 | *n/a* | 55.6-56-5 | 47.7-49.1 |
| GLR | *n/a* | *n/a* | *n/a* | 54.5 | 44.5 |
| P-Eucl | $46.45_{\pm4.03}$ | $67.54_{\pm3.54}$ | $71.45_{\pm2.98}$ | $43.15_{\pm3.12}$ | $32.56_{\pm3.68}$ |
| P-Hybrid | $47.87_{\pm2.03}$ | $72.48_{\pm4.57}$ | $72.87_{\pm1.75}$ | $46.85_{\pm2.17}$ | $36.42_{\pm4.08}$ |
| P-Poinc | $57.31_{\pm4.27}$ | $81.86_{\pm4.26}$ | $79.78_{\pm3.21}$ | $51.71_{\pm3.01}$ | $42.16_{\pm3.45}$ |

We run simulations for different manifold dimensions (ranging from $m = 2$ to $m = 12$) and pick the best one using the mean (across all 10 folds) cross-validation accuracy as criterion. We report the mean and standard deviation for the best $m$ on Table 1. We observe that the performance of the P-Eucl method is poor but the addition of the learnable parameterization (i.e., P-Hybrid method) leads to small but consistent improvement. The best performance is achieved by the fully hyperbolic representation (P-Poinc) and is on par or exceeds the performance of the state of the art. This suggests that the hyperbolic representation is vital for obtaining good performance. Additionally, the baselines have poor performance irrespectively of whether or not essential features have been included via the histogram of their birth times. This indicates that treating essential features separately and including them as additional inputs is not sufficient for good performance. These results support our initial motivation for representing persistence diagrams on the Poincare ball. Also, observe that our method out-performs all the competitor methods for the IMDB datasets, which have a small number of nodes and edges (approx. 10-20 nodes and 60-90 edges) while the REDDIT ones are bigger networks (approx. 400-500 nodes and 500-600 edges). This may indicate that the Poincare representation is best suited for smaller, less complex graphs.

## 4.2 IMAGE CLASSIFICATION

Even though it is well known that convolutional neural networks have achieved unprecedented success as feature extractors for images, we present an image classification case-study using topological features as a proof-of-concept for our method. Contrary to graphs, images are not inherently equipped with the structure of a simplicial complex. In theory, we could construct Vietoris-Rips complexes, as in Eq. 13, by interpreting pixels as a point cloud. However, this is not the most natural representation of an image, which has a grid structure. Therefore, we exploit this structure by constructing *cubical complexes*, i.e., unions of cubes aligned on a 2D grid. As in the case of graphs, we use two methods. For the first method, called *cubical filtration*, we use the grey-scale image directly and represent each pixel as a $2-$cube. Then, all of its faces are added to the $K$-th complex. We obtain a sublevel function by extending the grey-scale value $I(v)$ of pixel a $v$ to all cubes in $K$ as follows:

$$f(\sigma) = \min_{\sigma \text{ face of } \tau} I(\tau), \sigma \in K. \tag{14}$$

In the second experiment, we consider the problem of image classification. We utilize two standardized datasets: the MNIST, which contains images of handwritten digits, and the Fashion-MNIST, which contains shape images of different types of garment (e.g., T-shirt, trouser, etc.). Each dataset contains a total of 70K (60K train, 10K validation, 10 folds) grey-scale images of size $28 \times 28$. Both datasets are balanced and categorized into 10 classes. Finally, a grey-scale filtration is obtained using Eq. 2. The second method is called *height filtration* and uses the binarized version of the original image. We define the height filtration by choosing a direction $v \in \mathbb{R}^2$ of unit norm and by assigning to each pixel $p$ of value 1 in the binarized image a new value equal to $\langle p, v \rangle$, i.e., the distance of pixel $p$ to the plane defined by $v$. This creates a new grey-scale image which is then fed to the aforementioned cubical filtration. We note that the height filtration deserves special attention in persistent homology

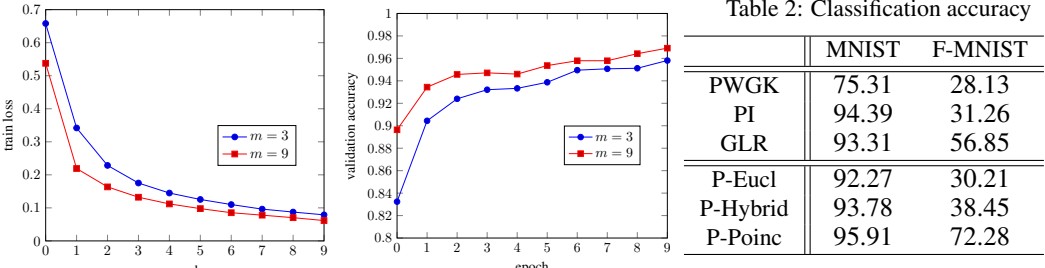

Table 2: Classification accuracy

|        | MNIST | F-MNIST |
|--------|-------|---------|
| PWGK   | 75.31 | 28.13   |
| PI     | 94.39 | 31.26   |
| GLR    | 93.31 | 56.85   |
| P-Eucl   | 92.27 | 30.21   |
| P-Hybrid | 93.78 | 38.45   |
| P-Poinc  | 95.91 | 72.28   |

Figure 3: Plotting the train loss (*left*) and the validation accuracy (*middle*) over 10 training epochs for the MNIST dataset using two different dimensions for the Poincare ball ($m = 3$ and $m = 9$).

because complexes to which it is applied can be approximately reconstructed from their persistence diagrams (4). In practice, we use direction vectors $v$ that are distributed uniformly across the unit cycle. We choose 30 and 50 directions for the MNIST and Fashion-MNIST, respectively.

We compare our method against three baselines that encompass all methods for handling persistence diagrams: (1) the Persistence Weighted Gaussian Kernel (PWGK) approach proposed by Kusano et al. (5), (2) the Persistence Images (PI) embedding developed by Adams et al. (10), and (3) the Learnable Representation based on Gaussian-like structure elements (GLR) by Hofer et al. (15). We run our simulations for manifold dimensions equal to $m = 3$ and $m = 9$ and report the best results in Table 2. Our method outperforms all other methods, in some cases by a considerable margin. We also study how the manifold dimension affects the train loss and validation accuracy. The results are shown in Fig. 3. Observe that for $m = 9$ the train loss decreases more rapidly and the validation accuracy increases more rapidly in the first few epochs. This suggests that a higher manifold dimension may be slightly better for representing persistence diagrams. Additionally, we observed that the Poincare representation tends to generalize better than its Euclidean counterpart. In fact, the validation accuracy of the P-Eucl method started decreasing after the first 2-3 epochs, whereas the P-Poinc method showed a saturation rather than a decrease. This was observed without modifying the dropout rate or any other hyper-parameters and is a strong empirical finding that demonstrates the superiority of Poincare representations.

## 5  CONCLUSION

We presented the first, to the best of our knowledge, method for learning representations of persistence diagrams in the Poincare ball. Our main motivation for introducing such method is that persistence diagrams often contain topological features of infinite persistence (i.e., essential features) the representational capacity of which may be bottlenecked when representing them in Euclidean spaces. This stems from the fact that Euclidean spaces cannot assign infinite distance to finite points. The main benefit of using the Poincare space is that by allowing the representations of essential features to get infinitesimally close to the boundary of the ball their distance to non-essential features approaches infinity, therefore preserving their relative importance. Directions for future work include the learning of filtration and/or scale end-to-end as well as to investigate whether or not there exists some hyperbolic trend in distances appearing in persistence diagrams that justify the improved performance especially on small graphs.

ACKNOWLEDGEMENTS

The authors gratefully acknowledge the support by the National Science Foundation under the Career Award CPS/CNS-1453860, the NSF award under Grant Numbers CCF-1837131, MCB-1936775, CNS-1932620, and CMMI 1936624 and the DARPA Young Faculty Award and DARPA Director's Fellowship Award, under Grant Number N66001-17-1-4044, and a Northrop Grumman grant. The views, opinions, and/or findings contained in this article are those of the authors and should not be interpreted as representing the official views or policies, either expressed or implied by the Defense Advanced Research Projects Agency, the Department of Defense or the National Science Foundation.

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

# A    HOMOLOGY AND DIFFERENTIAL GEOMETRY

## A.1    HOMOLOGY

Homology is a general method for associating a chain of algebraic objects (e.g abelian groups) to other mathematical objects such as topological spaces. The construction begins with a *chain group*, $C_n$, whose elements are the $n$-chains, which for a given complex are formal sums of the $n$-dimensional cells. The *boundary homomorphism*, $\partial_n : C_n \to C_{n+1}$, maps each $n$-chain to the sum of the $(n-1)$-dimensional faces of its $n$-cell, which is a $(n-1)$-chain. Combining the chain groups and the boundary maps in the sequence, we get the chain complex:

$$\ldots \xrightarrow{\partial_{n+2}} C_{n+1} \xrightarrow{\partial_{n+1}} C_n \xrightarrow{\partial_n} \ldots \xrightarrow{\partial_2} C_1 \xrightarrow{\partial_1} C_0 \xrightarrow{\partial_0} 0. \tag{15}$$

The kernels, $Z_n = ker(\partial_n)$, and the images, $B_n = im(\partial_{n+1})$, of the boundary homomorphisms are called *cycle* and *boundary* groups, respectively. A fundamental property of the boundary homomorphism is that its square is zero, $\partial_n \partial_{n+1} = 0$, which implies that, for every $n$, the boundary forms a subgroup of the cycle, i.e., $im(\partial_{n+1}) \subseteq ker(\partial_n)$. Therefore, we can create their quotient group

$$H_n := ker(\partial_n))/im(\partial_{n+1}) = Z_n/B_n, \tag{16}$$

which is called $n$-*th homology group* and its elements are called *homology classes*. The rank, $\beta_n = \text{rank}(H_n)$, of this group is known as the $n$-*th Betti number*.

Even though homology can be applied to any topological space, we focus on *simplicial homology*, i.e., homology groups generated by simplicial complexes. Let $K$ be a simplicial complex and $K_n$ its $n$-skeleton. The chain group $C_n(K)$ is the free abelian group whose generators are the $n$-dimensional simplexes of $K$. For a simplex $\sigma = [x_0, x_n] \in K_n$, we define the boundary homomorphism as

$$\partial_n(\sigma) = \sum_{i=0}^{n}[x_0, \ldots, x_{i-1}, x_{i+1}, \ldots, x_n] \tag{17}$$

and linearly extend this to $C_n(K)$, i.e., $\partial_n(\sum \sigma_i) = \sum \partial(\sigma_i)$.

## A.2    RIEMMANIAN MANIFOLDS

An $m$-*dimensional manifold* can be seen as a generalization of a 2D surface and is a space that can be locally approximated by $\mathbb{R}^m$. The space in which the manifold is embedded is called *ambient space*. We assume that the ambient space is the standard Euclidean space of dimension equal to the dimension of the manifold, i.e, $\mathbb{R}^m$. For $x \in \mathcal{M}$, the *tangent space* $T_x\mathcal{M}$ of $\mathcal{M}$ at $x$ is the linear vector space spanned by the $m$ linear independent vectors tangent to the curves passing through $x$. A *Riemannian metric* $g = (g_x)_{x \in \mathcal{M}}$ on $\mathcal{M}$ is a collection of inner products $g_x : T_x\mathcal{M} \times T_x\mathcal{M} \to \mathbb{R}$ varying smoothly across $\mathcal{M}$ and allows us to define geometric notions such as angles and the length of a curve. A *Riemannian manifold* is a smooth manifold equipped with a Riemannian metric $g$. The Riemannian metric $g$ gives rise to the metric norm $\|\cdot\|_g$ and induces global distances by integrating the length of a shortest path between two points $x, y \in \mathcal{M}$

$$d(x, y) = \inf_{\gamma} \int_0^1 \sqrt{g_{\gamma(t)}(\dot{\gamma}(t), \dot{\gamma}(t))} dt, \tag{18}$$

where $\gamma \in \mathcal{M}^\infty([0, 1], \mathcal{H})$ is such that $\gamma(0) = x$ and $\gamma(1) = y$. The unique smooth path $\gamma$ of minimal length between two points $x$ and $y$ is called a *geodesic* and the underlying shortest path length is called *geodesic distance*. Geodesics and can be seen as the generalization of straight-lines. For a point $x \in \mathcal{M}$ and a vector $v \in T_x\mathcal{M}$, let $\gamma_v$ be the geodesic starting from $x$ with an initial tangent vector equal to $v$, i.e., $\gamma_v(0) = x$ and $\dot{\gamma}_v(0) = v$. The existence, uniqueness and differentiability of the exponential map are guaranteed by the classical existence and uniqueness theorem of ordinary differential equations and Gauss's Lemma of Riemannian geometry (32).

## A.3    POINCARE BALL

We define the Möbius addition for any $x, y$ on the Poincare ball:

$$x \oplus y = \frac{(1 + 2\langle x, y \rangle + \|y\|^2)x + (1 - \|x\|^2)y}{1 + 2\langle x, y \rangle + \|x\|^2 \|y\|^2} \tag{19}$$

The Möbius addition is a manifold preserving operator that allows us to add point belonging in the Poincare ball without leaving the space. Finally, we provide the coordinate chart $\psi : \mathcal{M} \to \mathbb{R}^m$, i.e., the inverse of the parameterization in Eq. 8, which is defined as follows:

$$x_i = x_1 \prod_{j=2}^{i-1} \sin(y_j) \cos(y_i), \text{ for } i = 1, \ldots, m-1 \text{ and } x_m = x_1 \prod_{j=2}^{m-1} \sin(y_j) \sin(y_m). \quad (20)$$

Note that we have omitted the learnable parameters. The coordinate chart is not explicitly needed for constructing the representation. However, it could be useful as an intermediate layer between our representation and standart deep neural network architecture. Its purpose would be to transfer the representation from the manifold to a Euclidean space which can be fed into a standart DNN without having to define its operations on the manifold.

## B  THEORETICAL RESULTS

*Proof of Theorem 1.* Let $\mathcal{D}, \mathcal{E}$ be two persistence diagrams and $\eta : \mathcal{D} \to \mathcal{E}$ bijection that achieves the infinum in Eq.5. Consider the subset $\mathcal{D}_0 = \mathcal{D} \setminus \mathbb{R}_\Delta$, which is essentially the original diagram without the points in the diagonal. Then, we have the following sequence of inequalities

$$d_{\mathcal{B}}(\Phi(\mathcal{D}, \theta), \Phi(\mathcal{E}, \theta)) \quad (21)$$

$$= d_{\mathcal{B}}\left( \exp_{x_0}\left( \sum_{x \in \mathcal{D}} \log_{x_0}\left( \phi(\rho(x)) \right) \right), \exp_{x_0}\left( \sum_{x \in \mathcal{E}} \log_{x_0}\left( \phi(\rho(x)) \right) \right) \right) \quad (22)$$

$$\leq K_e \left\| \sum_{x \in \mathcal{D}} \log_{x_0}(\phi(\rho(x))) - \sum_{x \in \mathcal{E}} \log_{x_0}(\phi(\rho(x))) \right\|_g \quad (23)$$

$$\leq K_e \left\| \sum_{x \in \mathcal{D}_0} \log_{x_0}(\phi(\rho(x))) - \sum_{x \in \mathcal{D}_0} \log_{x_0}(\phi(\rho(\eta(x)))) \right\|_g \quad (24)$$

$$\leq K_e \sum_{x \in \mathcal{D}_0} \left\| \log_{x_0}(\phi(\rho(x))) - \log_{x_0}(\phi(\rho(\eta(x)))) \right\|_g \quad (25)$$

$$\leq K_e K_l \sum_{x \in \mathcal{D}_0} \left\| \phi(\rho(x)) - \phi(\rho(\eta(x))) \right\|_g \leq K_e K_l K_\phi \sum_{x \in \mathcal{D}_0} \left\| \rho(x) - \rho(\eta(x)) \right\|_g \quad (26)$$

$$\leq K_e K_l K_\phi K_\rho \sum_{x \in \mathcal{D}_0} \left\| x - \eta(x) \right\|_g \leq K_e K_l K_\phi K_\rho w_1^g(\mathcal{D}_0, \mathcal{E}) \quad (27)$$

where $w_1^g$ is the Wasserstein distance with the $q$-norm replaced with the norm induced by the metric tensor of the Poincare ball. Eq. 23 follows from the smoothness of the exponential map (assuming that it has Lipschitz constant equal to $K_e$). Eq. 24 follows by substituting the bijection $\eta$ and because, as per requirements of Theorem 1, the auxiliary transformation $\rho$ is zero on the diagonal $\mathbb{R}_\Delta$ and $x_0 = 0$. Note that the requirements of Theorem 1, combined with Eq. 8, 10 and 19, imply that the summation over $x \in \mathbb{R}_\Delta$ collapses to zero. Eq. 25 follows from the triangle inequality. Eq. 26 follows from the smoothness of the logarithmic map and the parameterization (assuming $K_l$ and $K_\phi$ Lipschitz constants, respectively). Finally, Eq. 27 follows from the assumed, as per Theorem 1, smoothness of $\rho$ and by using the definition of Wasserstein distance. This concludes the proof. $\square$

## C  IMPLEMENTATION DETAILS

**Auxiliary Transformation**: Even though several choices for the auxiliary transformation $\rho : \mathbb{R}^2_* \to \mathbb{R}^m$ defined in Eq. 7 are possible, we present the one used in our implementation. Our choice for $\rho$ is based on the exponential structure elements introduced by Hofer et al. (15). Formally, for $i = 1, 2, \ldots m$, we define the following

$$\rho_i(x_1, x_2) = \begin{cases} e^{-(x_1 - \mu_{1,i})^2 - (\ln(x_1 - x_2) - \mu_{2,i})^2} & x_1 \neq x_2 \\ 0 & x_1 = x_2. \end{cases} \quad (28)$$

Then, the auxiliary transformation is $\rho : (x_1, x_2) \rightarrow (\rho_i(x_1, x_2))_{i=1}^m$, where $(\mu_{1,i}, \mu_{2,i})_{i=1}^m$ are the means. In contrast to Hofer et al. (15), those means are not learned, but we rather fix them to pre-defined values and keep then constant during training. Additionally, we can easily prove that the above-defined $\rho$ satisfies the stability conditions given by Theorem 1, i.e., zero on the diagonal and Lipschitz continuity. Finally, under the assumption of all means being unique, it is injective on $\mathbb{R}_*^2$ and, therefore, introduces no information loss.

**Network Architecture**: We implemented all algorithms in TensorFlow 2.2 using the TDA-Toolkit[2] and the Scikit-TDA[3] for extracting persistence diagrams and run all experiments on the Google Cloud AI Platform. To show the versatility of our approach, we used the same network architecture across all of our simulations and our proposed representation acts as an input. In more detail, for each individual filtration that we extract from the data (images or graphs), we create a different input layer. The input to this layer is the persistence diagrams of all homology dimension given by the corresponding filtration. Given the nature of the data (graphs and images), the homology dimensions with non-trivial persistence diagrams are only the first two (i.e., $H_0$ and $H_1$). Our input layer processes the persistence diagrams of each homology class independently and outputs the representations (i.e., Eq. 11) for each one of them. Following that, we concatenate and flatten the previous outputs and fed the resulting vector to a batch normalization layer. Then, we use two dense layers of 256 and 128 neurons, respectively, with a Rectified Linear (ReLu) activation function. This is followed by a dropout layer to prevent overfitting and a final dense layer as an output. We consider two different implementations of the the above architecture. The first one is based on the hyperbolic neural networks introduced by Ganea et al. in (22). It re-defines key neural network operations in the Poincare ball and, therefore, allows us to directly utilize the output of our representation in the neural network. The second implementation uses the coordinate chart given by Eq. 20 to project the representation back to a Euclidean space and feed it to a standardized implementation of the aforementioned layers. This method is more convenient in practice as it maintains compatibility with existing backbone networks trained in Euclidean geometry.

**Hyperparameters and Training**: To train the neural network, we use the Adam optimizer ($\beta_1 = 0.9, \beta_2 = 0.999$) with an initial learning rate of 0.001 and batch size equal to 64. We use a random uniform initializer in the interval $[-0.05, 0.05]$ for all learnable variables. For all graph datasets we trained the network for 100 epochs and halved the learning rate every 25 epochs. For the MNIST and Fashion-MNIST datasets we used 10 and 20 epochs, respectively, and no learning rate scheduler. We tune the dropout rate manually by starting from really low values and monitoring the validation set accuracy. In general, we noticed that the network did not tend to overfit, therefore, we kept the rate at low values (0-0.2) for all experiments.

---

[2]`https://github.com/giotto-ai/giotto-tda`
[3]`https://github.com/scikit-tda/scikit-tda`

