# OpenReview forum: "Learning Hyperbolic Representations of Topological Features"
_ICLR.cc/2021/Conference — ICLR 2021 Poster_

### Official Review · AnonReviewer2 · 2020-10-28
**A novel embedding method for persistence diagrams**

**Rating:** 7
**Confidence:** 5

**Review:**

# Synopsis of the paper

This paper proposes a novel embedding or representation algorithm for
persistence diagrams, i.e. topological descriptors. Existing methods are
restricted because they do not feature *learnable* or *trainable*
parameters for their representations (with the exception of methods such
as the one by Hofer et al. or PersLay). This method, by contrast,
presents a trainable embedding to the Poincaré ball, thus representing
persistence diagrams in a hyperbolic space.

This has the advantage of being more appropriate for representing
*essential features*, i.e. features of infinite persistence. Thus,
it is possible to perform end-to-end training of neural networks
that use persistence diagrams as their input.

Experiments with graph classification and image classification tasks
demonstrate the utility of the proposed method.

# Summary of the review

This is a very well-written paper, with a strong contribution to
topological data analysis and machine learning. The paper is technically
sound, apart from some minor inconsistencies, which I shall discuss
below. It is exciting for me to see how to obtain fully-trainable
embeddings here, and I am happy to endorse this paper.

There are a few issue that I would like to see rectified in a revision
of the paper, though:

1. Some details on the method are missing. This concerns primarily the
   learnable auxiliary transformation from Eq. 7. While the paper
   subsequently discusses potential choices for this function, this is
   not specified. In fact, the method tying everything together even
   cites this equation again, without providing a definition for it.

   This needs to be rectified---I am assuming that one could, for
   example, use a transformation such as the one provided by Hofer et
   al.; is this correct?

2. The experiments on graph classification are lacking some comparison
   partners. Specifically when discussing topology-based approaches, it
   is useful to compare to other topology-based approaches. Here are
   some suggestions:

      - Hofer et al.: *Graph Filtration Learning*
      - Rieck et al.: *A Persistent Weisfeiler–Lehman Procedure for Graph Classification*
      - Zhao & Wang: *Learning metrics for persistence-based summaries and applications for graph classification*

    There is an overlap of the data sets assessed using these methods,
    so the appropriate results could be cited and used. I wanted to
    raise this concern primarily because I am aware of other methods
    obtaining somewhat better classification performance in some cases.

    It would strengthen the paper immensely if it could perform some
    form of 'ablation study', i.e. highlighting to what extent the
    improved results are driven by the embedding in hyperbolic space, or
    the choice of filtration, etc.

3. Adding to this, some details of the method need to be described
   better; only the supplements briefly mention how $m$, the embedding
   parameter, is chosen in the end, but I would prefer a more in-depth
   analysis of the choice of this parameter. Is it sufficient to pick,
   say, $m = 32$ for all practical purposes? Or does it make sense to
   concatenate the representations afterwards, as discussed in the
   supplements? Adding more details will aid in understanding the paper
   and, ultimately, promote further adoption of the method.

# Detailed comments

- In the abstract, I would say that 'Existing methods are restricted in
  terms of their expressivity'. I would use 'bottleneck' only to refer
  to the eponymous distance, in order to prevent confusion.

- The definition of 'filtration' in the introduction is incorrect;
  a filtration is a *sequence* of subspaces.

- I would not refer to diagrams being *injective*. I understand what is
  mean here, viz. the fact that under some assumptions (!), one can
  reconstruct the input space from a diagram (this is known as solving
  the inverse problem), but I think this might be slightly confusion
  here. Why not focus on the expressivity properties of diagrams and the
  fact that one can *approximate* their inputs under some conditions?

- The comment on 'extended persistence' on p. 2 is slightly imprecise;
  'extended persistence' is well studied now, but PersLay is indeed the
  first 'deep learning' method incorporating it. This should be
  clarified.

- When discussing filtrations and homology groups, I would stick with
  $\subseteq$ instead of $\subset$. The former is more generic and would
  make the write-up more consistent, as currently both forms are being
  mixed.

- As outlined above, I would suggest to be more verbose when it comes to
  the description of the method in terms of the individual functions,
  i.e. $\psi$ and $\rho$.

- Am I mistaken or could $\rho$ also be *any* universal approximator,
  such as a set function or, more generically, a deep neural network?
  Would it be possible to use the method by Hofer et al. (2017) to
  obtain this map and *then* subsequently train a better hyperbolic
  embedding?

- When running the experiments, is $m$ fixed or is the best $m$ selected?
  I see that $m \in \{2, \dots, 12\}$, but I do not understand how this
  is actually used in the network. Figure 4 is somewhat helpful here,
  but it would be interesting to see what happens for a range of
  parameters.

- It is my understanding that the Euclidean method P-Eucl is only driven
  by $\rho$. Is this correct? How critical is this choice in practice?
  Moreover, how was it chosen for this paper? I am asking because it is
  clear that the hyperbolic embedding helps improve predictive
  performance, but I wonder what would happen if one uses an
  'underpowered' $\rho$ function.

All in all, I feel that this could make a very strong addition to the
topological machine learning literature!

# Style & clarity

The paper is very well-written, the authors are to be commended for
that. I have a few minor suggestions, some of which are more personal
pet peeves, but which might help make this paper shine even more!

- 'root of a complex polynomial' --> 'roots of a complex polynomial' (?)

- I would prefer not to use citations as nouns, i.e. I would prefer
  writing 'Kusano et al. (5)' instead of 'In (5), Kusano et al.'; the
  former strikes me as more readable and also translates well to
  different citation styles.

- The image of $f_n^{i,j}$ is *the* $n$th persistent homology group

- Whenever possible, I would use $\operatorname{text}$ for operators or
  functions, instead of 'raw' $text$. This concerns for example the rank
  function but also the degree function.

- I would write 'Poincaré' everywhere

- 'persistent diagram' --> 'persistence diagram'

- 'persistent image' --> 'persistence image'

- 'Vietories' --> 'Vietoris'

- 'gray-scale' --> 'grey-scale' (or vice versa, if American English is
  the preferred spelling)

---

> ### Author Response · Authors · 2020-11-17
> **Response to Reviewer Feedback**
>
> We would like to thank the reviewer for his/her comprehensive feedback and we are glad that s/he liked our paper. We found your suggestions very insightful and they helped us bring the paper in much better shape.
>
> # Summary of the review - Response
>
> 1) That is correct, we can use any of the representations developed by Hofer. In fact, there is not one single choice for $\rho$. To guarantee stability, there are two conditions that $\rho$ needs to satisfy and we have re-written them more clearly in Theorem 1. To ensure that the representation is not only stable but also expressive, we require injectivity (please also see response to your comment about an “underpowered” $\rho$). We have added a relevant discussion below Theorem 1.  The exact formula for $\rho$ is of little interest and we have added it in the implementation details in the appendix.
>
> 2) Thank you for providing the references. Indeed the GFL and the WKPI methods achieve better performance on some datasets that the state of the art we had included and we added them. The P-WL focuses on labeled graphs, which our method cannot currently handle, so direct comparison is difficult. Additionally, we adopted your suggestion about the ablation study and added a paragraph in the experimental section. The P-Eucl and P-Poinc variants are essentially an ablation study and we added a hybrid method (H-Hybrid). Please have a look at that part and the newly added experimental results. We believe that you will find them interesting.
>
> 3) We moved that part from the appendix to the experimental section of the paper and elaborated on how we select the manifold dimension and the number ($K$) of the so-called projection bases (i.e., the replicas which we concatenate). Both of these are treated as hyper-parameters and the empirical rule for picking them is as follows: We initially attempt small values for the manifold dimension (e.g, $m=3$) and pick the number of projection bases depending on the type of data (i.e, in the range 5-10 for images and 200-500 for graphs). As we increase m, we decrease K to maintain a similar model capacity and avoid overfitting. We cannot say that a certain manifold dimension (e.g., $m=32$) would suffice for all practical cases, as it could differ from one dataset to another. In practice, we choose the optimal combination via the validation dataset. There is a discussion about how increasing m affects the validation performance at the end of Sec. 4.2.
>
> # Detailed comments - Response
>
> >In the abstract, I would say that 'Existing methods are restricted in terms of their expressivity'. I would use 'bottleneck' only to refer to the eponymous distance, in order to prevent confusion.
>
> Thank you for this suggestion. We like  the term “expressivity” and we adopted it throughout the paper whenever appropriate.
>
> > The definition of 'filtration' in the introduction is incorrect; a filtration is a sequence of subspaces.
>
> We corrected this.
>
> > I would not refer to diagrams being injective. I understand what is mean here, viz. the fact that under some assumptions (!), one can reconstruct the input space from a diagram (this is known as solving the inverse problem), but I think this might be slightly confusion here. Why not focus on the expressivity properties of diagrams and the fact that one can approximate their inputs under some conditions?
>
> You are right, “injective” is misleading. We are referring to the inverse problem, as you noted. We re-phrased to avoid confusion.
>
> >The comment on 'extended persistence' on p. 2 is slightly imprecise; 'extended persistence' is well studied now, but PersLay is indeed the first 'deep learning' method incorporating it. This should be clarified.
>
> We re-phrased this to emphasize that PersLay is the first deep learning method to utilize extending persistence.
>
> > When discussing filtrations and homology groups, I would stick with ⊆ instead of ⊂. The former is more generic and would make the write-up more consistent, as currently both forms are being mixed.
>
> We corrected this.
>
> >As outlined above, I would suggest to be more verbose when it comes to the description of the method in terms of the individual functions, i.e. ψ and ρ.
>
> We tried to be more verbose on what $\rho$ is and the conditions that it needs to satisfy. Regarding $\psi$, we chose not to add any details in the main text as they are of little interest for our method and we prefer to limit the number of equations that the reader is exposed to. The analytical expression for it is still given in the appendix.
>
> (continued)

---

> > ### Comment · AnonReviewer2 · 2020-11-24
> > **Thanks!**
> >
> > Thanks a lot for your detailed response and the large number of changes you implemented, specifically those that pertain to an improved 'disentanglement' of the individual constituents of your method. The baseline values look encouraging to me, and I appreciate the time you took to rewrite and extend the paper!
> >
> > I have one tiny suggestion left to make: the reference for GFL appears to be outdated; please see the [PMLR page](http://proceedings.mlr.press/v119/hofer20b.html) for the most recent one (the paper was presented at this year's ICML conference, AFAIK).

---

> > > ### Author Response · Authors · 2020-11-24
> > > **Thank you!**
> > >
> > > We are glad that you are satisfied with our revision and the added baselines!
> > >
> > > We have updated the reference for the GFL paper.

---

> ### Author Response · Authors · 2020-11-17
> **Response to Reviewer Feedback**
>
> (continued)
>
> >Am I mistaken or could ρ  also be any universal approximator, such as a set function or, more generically, a deep neural network? Would it be possible to use the method by Hofer et al. (2017) to obtain this map and then subsequently train a better hyperbolic embedding?
>
> In practice, this auxiliary $\rho$ could be any generic neural network approximator or the representation method by Hofer et al. (2017) or the PersLay by Carriere et al. (2020). Note that such a $\rho$ would not satisfy, in general, the conditions of stability given by Theorem 1. Nonetheless, in practice, even unstable representations seemed to perform equally good in the classification tasks. In fact, we even tried picking $\rho$ to be a simple zero-padding embedding, which does not satisfy the second condition of Theorem 1, i.e, it is not zero on the diagonal. The classification performance was equally good. This is explainable because the condition requiring $\rho$ to be zero on the diagonal exists due to theoretical artifacts. Please see the discussion following Theorem 1.
>
> >It is my understanding that the Euclidean method P-Eucl is only driven by ρ. Is this correct? How critical is this choice in practice? Moreover, how was it chosen for this paper? I am asking because it is clear that the hyperbolic embedding helps improve predictive performance, but I wonder what would happen if one uses an 'underpowered' ρ function.
>
> In all cases (P-Eucl, the newly added P-Hybrid and the P-Poinc), \rho is the same (as described in the newly added paragraph in the Appendix C). To obtain the P-Hybrid, we need to replace the Poincare ball with the Euclidean space. This implies that the exponential and logarithmic maps reduce to identities. Additionally, we obtain P-Eucl  from P-Hybrid by replacing the learnable parameterization given by Eq. 8 by simple addition of the learnable parameters. Please see the ablation study paragraph in the beginning of the experimental section.
>
> Your comment about an “underpowered” $\rho$ function highlights an important subtlety: The stability theorem (Theorem 1) does not prohibit us from choosing a degenerate $\rho$. For example, $\rho=0$ satisfies the conditions of Theorem 1 and therefore leads to a stable representation. However, such representation is obviously not useful for learning tasks. An implicit requirement for $\rho$ is that it is injective (everywhere except the diagonal), which, given that $\rho$ is a higher dimensional embedding this is a mild condition, fairly easy to satisfy. Please see the discussion following Theorem 1.
>
> # Style and clarity - Response
> We correct typos and/or adopted your suggestions.

---

> > ### Comment · AnonReviewer2 · 2020-11-23
> > **Thanks**
> >
> > Thanks; this is an excellent clarification of my original points! I'll add another comment to your other response soon.

---

### Official Review · AnonReviewer3 · 2020-10-28
**Learning Hyperbolic Representations of Topological Features**

**Rating:** 6
**Confidence:** 4

**Review:**

The authors propose to learn a representation for the persistence diagram (PD) in the hyperbolic space to incorporate the essential features (i.e., infinite persistence). The authors show that the hyperbolic representation has stability. Empirically, the authors illustrate that the hyperbolic representation for PD compares favorably with other baselines on graph and image classification.

The motivation to use hyperbolic representation to include essential features for PD is interesting. The authors give some details about the background (e.g., persistence diagrams, Poincare ball). However, the main part of the framework and how to learn the parameters of the embedding are missing.

As I understand, the authors propose:

(1) to use some auxiliary transformation to lift points of $R^2$ into $R^m$ (however, there are no properties or information about this auxiliary transformation, and how one can do it)

(2) projects points in $R^m$ into the Poincare ball, parameterized by $\theta$. Why ones need this parameterization, and why this parameterization is important in applications? (It seems the authors combine projection with some transformation here?). It is better in case the authors give the projection (w.r.t. what distance?) and then transform the projected points.

+ There is no description of the space of $\Theta$ for the parameterization?

(3) combine the representations of each point in PD. It seems that the authors use the sum of all points w.r.t. hyperbolic manifold to represent PD.

+ I also concern about the usage of exp and log map at point $x$ for this combination. Typically, the tangent space is just simply a "flattened" space of the hyperbolic space at point $x$, it can only preserve the geometry for some close neighbor points of $x$ (this "flat" approach has a large distortion for those points which are far to $x$. It is better in case the authors give more discussion about how to choose $x$, and how it affects the geometry of the hyperbolic?

The main framework to learn those parameters is not presented in the main manuscript, and it is also unclear how one uses the learned hyperbolic representation for the downstream task?
+ It seems that the authors incorporate the procedure to learn those parameters inside the networks for classification and learn end-to-end?

Somehow, it seems that the authors propose to use a hyperbolic embedding within a neural network as a classifier for PDs. (in my opinion, the novelty may be incremental in case the authors simply replace the Gaussian-like embedding of Hofer et al. into a hyperbolic embedding inside a neural network. However, the hyperbolic representation for PD may be still interesting by itself. )

Overall, I think this work seems interesting and has good potential. The authors may need to describe more details about the neural framework use to learn the parameters of the hyperbolic representation.

---

> ### Author Response · Authors · 2020-11-17
> **Response to Reviewer Feedback**
>
> We would like to thank the reviewer for the feedback and we are glad that s/he found hyperbolic representations of persistence diagrams an interesting research topic. Based on the feedback, it seems that the reviewer has fully understood our method. We provide clarifications on details that may have been unclear in our paper.
>
> 1) Regarding the properties of the auxiliary transformation, we kindly point the reviewer to Theorem 1. We have re-written the theorem to highlight the properties. The auxiliary transformation \rho is assumed to be Lipschitz continuous and zero on the diagonal, i.e., \rho(x) = 0 for all x \in \mathbb{R}_\Delta. Both assumptions are needed to ensure stability as per Theorem 1. There is a short explanation of why we need \rho(x) = 0 on the diagonal and the full proof is provided in Appendix B. Please also note that we have added (following comments from Reviewer 2) a paragraph after Theorem 1 where we explain another property of \rho.
>
> 2) The “projects” word is a misleading choice. We use it interchangeably with “transform”. What we are actually doing is transforming the points from \mathbb{R}^m to the Poincare ball via the learnable parameterization given by Eq. 8. We rephrase the text replacing the word “project” with “transform” to avoid any confusion. The parameter space \Theta is essentially the R^m space, we corrected that. The parameters \theta are a pivotal part of our method because they allow us to learn the representation of each point in the PD (and consequently, to learn the representation of the PD itself) on the Poincare ball. Our method is somehow similar to the one by Hofer et al. in the sense that both learn the representations of PD and feed them to neural networks for classification tasks. However, we believe that this is not an incremental contribution; as the reviewer points out, the hyperbolic representation of PDs is the main novelty of our work.
>
> 3) We kindly point the reviewer to Theorem 1 where we state the choice of x (or x_0 as denoted in that theorem). We choose x_0=0 which guarantees that the representation is stable. As before, this is discussed in the paragraph following Theorem 1. The choice of x_0 does not have any major impact on our method. Please note that the desired property of the Poincare space (i.e., the ability to assign infinite distances to finite points) holds irrespectively of the chosen x_0. The exponential and logarithmic maps are utilized so that we can combine the representations of individual points of the PD into a single point in the Poincare space. Regarding your comment about the tangent space. The tangent space is a vector space and if we equip it with the Euclidean metric it does indeed cause distortion on geometric quantities. However, we typically equip the tangent space with the induced metric.  The transformations given by these maps are norm-preserving, i.e., for example, the geodesic distance from $x$ to the transformed point $\exp_{x}(v)$ coincides with the metric norm $||v||_g$ induced by the metric tensor $g^\mathcal{B}_x$.
>
> >It seems that the authors incorporate the procedure to learn those parameters inside the networks for classification and learn end-to-end?
>
> This is in fact exactly what we are doing. The learnable representation acts as an input layer in a DNN and the parameters are learned end-to-end. The reviewer is right, we are not explicitly mentioning that in the main text, we only implicitly mention it in Appendix C. We added a short clarification in the beginning of the experiments section (Sec. 4). For reason for mentioning this in the experiment section rather that Sec. 3 is the following: Even though our method is validated on a neural network classifier using the representation as an input layer, the hyperbolic representation itself may be of independent interest, as the reviewer points out, not necessarily tied to the fact that is used as input to a neural network or to the classification task.

---

> > ### Comment · AnonReviewer3 · 2020-11-24
> > **Thanks for the clarification**
> >
> > Thank you for your clarification.
> > I am happy to increase my rating for the submission (5-->6).

---

### Official Review · AnonReviewer4 · 2020-10-28
**Interesting direction but needs to be motivated and discussed more**

**Rating:** 6
**Confidence:** 3

**Review:**

The authors propose to learn representations of topological persistence diagrams in hyperbolic spaces (Poincare balls). They provide a step-by-step methodology, and an algorithm for creating the representation. They also compare their approach with various graph classification and other ways of representing persistence diagrams.

Strengths:
=========
1. This is the first work that learns hyperbolic representations of persistence diagrams.
2. The methodology is sufficiently clear and the paper is reasonably well-written.
3. Applications to graph data as well as image data show some promise.

Things to improve:
=================
1. My biggest question in this paper is around the assumptions the authors have made and the motivation. They have mentioned that essential features have infinite persistence and truncating the persistence is not the best thing to do. I agree, however, there is really only one essential feature - the 0th order persistent homology group that becomes a single connected component at a scale that is sufficiently large. Since this "essential" group is one and the same for all point clouds that we compare,  they always match to one another, and it is safe to say that this truly essential group can be ignored in ideal cases. All other "essential" groups are only artifacts of our construction, which happen because we do not use a large enough scale for computational or some other reason.

2. Even in an ideal setting (with large enough scales), it would be good to discuss why hyperbolic coordinates make sense. Why do we expect the distance between the persistence homology module to have some hyperbolic trend (see fig. 2 in https://dawn.cs.stanford.edu/2018/03/19/hyperbolics/). Some thought and discussion around this would be helpful to understand the motivation of the approach.

3. Based on fig. 2 I was initially under the impression that individual points will be mappable between persistence diagrams and hyperbolic representations, but it took some more reading to understand that there is one single representation for the entire persistence diagram in the hyperbolic space. Perhaps the authors can make it clear somewhere (even in Fig. 2).

4. How is x0 chosen in (11). Why is summing in tangent space meaningful? Why not some  other operation?

5. What is the absolute state-of-the-art  for Table 1 and Table 2? I am not holding this as a negative, but it may be good for the readers to know where TDA methods stand in these applications.

6. Along similar lines, are there any applications that are especially suited to TDA that the authors can demonstrate?

7. The authors seem to have used multiple runs for results in Table 1, but the text in Sec. 4.1 does not indicate how it is done. Please elaborate.

8. Why are some values missing for some methods in Table 1.

9. Why does P-Eucl make sense as a baseline? Is this like a so-called ablation study for this procedure? How are the representations created for P-Eucl? Please feel free to add more details on this in supplement.

10. In Sec. 4.2, do the  compared methods also use the same simplicial construction (cubical complexes constructed in a  specific way)? If so, would the compared baselines work better with some other construction?

In summary, this is an interesting direction, but clarity is needed regarding the motivation,  and presenting extra details along the way will also be appreciated.  Finally, it could make sense to think about  an appropriate application.

Update post-rebuttal
==================
- I am happy with the thorough engagement by the authors and their clarifications. So  I am bumping the up the score a notch.

---

> ### Author Response · Authors · 2020-11-17
> **Response to Reviewer Feedback**
>
> We would like to thank the reviewer for the feedback. We give our responses to your comments bellow:
>
> 1) The scale is indeed very important for determining when features appear and when/if they disappear. But the filtration plays a pivotal role as well. Consider for example a filtration over graphs where a certain loop appears, unaltered, at the very beginning and the very end of the filtration. The persistence of this loop (i.e., 1-dim homology group) is infinite as it never “dies” in the filtration. We kindly ask you to look at the section “Extended Persistence Diagrams” of the paper by Carriere et al. (ref. 19 in our paper) where that same example is mentioned and used for motivating the introduction of extended persistence in the deep learning context.
> 2) The hyperbolic representation is meaningful as it allows us to treat essential and non-essential features via a unified representation. The Euclidean representation would shrink the relative importance of essential features and the Euclidean metric cannot assign infinite distance to finite points.
> 3) Yes you are right, that figure might give that impression. We added a label to emphasize that the PD is mapped on a single point in the Poincare ball.
> 4) The choice of $x_0$ is highlighted in Theorem 1. We choose $x_0=0$ to guarantee the stability of the representation.
> 5) It is hard to pick an absolute state of the art as different methods perform differently across different datasets. We have included more state of the art results in Table 1 to show the comparison with an even wider range of methods.
> 6) The method is not necessarily best suited for any special application. It is a generic method for learning representations of persistence diagrams extracted from any type of data to which TDA can be applied (e.g., graphs, images, time-series, high-dimensional point clouds etc).
> 7) The multiple runs are the different cross-validation folds. We have highlighted that.
> 8) Some values are not available for the corresponding dataset in the respective papers.
> 9) In fact, the P-Eucl does act as an ablation study for this procedure. We have added a paragraph in the experimental section elaborating on P-Eucl and a  newly added P-Hybrid method.
> 10) For the baselines, we used the same PDs as the ones we used for our method. We obtained the baseline results using the publicly available code from the respective paper.

---

> > ### Comment · AnonReviewer4 · 2020-11-22
> > **Response clarifies some questions**
> >
> > Thanks to the authors for their response.
> >
> > I now understand that filtration plays a pivotal role even though in an absolute sense there is really only one essential feature. Like the authors pointed out Carriere et al. (ref. 19 in the paper) provides some hints around this. The authors should attempt to  clarify this in the  text, since Reviewer 1 also seems to have a similar question (For graph with Rips filtration, why are there 1-dim essential homology? Wouldn't all 1D homology features be killed eventually?). I am still not  completely convinced that we  should treat the points with apparently infinite persistence this way (the  extended persistence seems to introduce a novelty in the construction of the diagrams rather than just in the interpretation like what the authors do here). However, I  am willing to accept that this is one possible way of doing this.
> >
> > Also, I do not see any changes in Fig. 2 compared to the initial revision that clarifies my original comment 3. Am I missing something?
> >
> > Looking at table 1, and the paragraph just above sec. 4.2, while you claim that P-Poinc is on par or exceeds state of the art, I see this only for 2  out of 5 datasets shown. I do not really expect any one method to outperform other TDA and non-TDA methods in all datasets.  This is why I was asking if the authors think there is  some niche application where  treating essential  features this way provides a good advantage (my original comment 6). It would be great if the authors consider this and provide their thoughts. This is also a good way to make this method more impactful in the crowded space of various distances between persistence diagrams.

---

> > > ### Author Response · Authors · 2020-11-24
> > > **Follow-up on reviewer's comments**
> > >
> > > Thank you for your second response.
> > >
> > > We now understand better what you meant by your initial comment.  We kindly ask you to have a quick look at this: (https://ripser.scikit-tda.org/notebooks/Basic%20Usage.html) practical computation of PDs under the VR filtration for a simple point cloud and focus on cells 9-12. As you can see, by varying the maximum radius of the VR filtration we obtain 0-th and (multiple) 1-th dimensional essential features. While we may be able to increase the scale sufficiently high and end up having only the trivial essential feature that you mention, the resulting persistence diagrams may not be the ones that best summarize the data in terms of performance on the underlying learning task. We have added this comment in the second paragraph on page 2.
> > >
> > > The process of obtaining the PD is essentially a feature extraction process and the scale can be considered as a hyper-parameter (among others).  In our experimental setup, we set all parameters related to PD extraction to pre-defined constants. Extending this to account for learnable parameters in the PD extraction process (such as parameters that appear in the filtration function or the scale itself) is an interesting direction for future work. We mention the paper Graph Filtration Learning by Hofer et al. (also mentioned by Reviewer 2) which is along these lines.
> > >
> > > Extended persistence is indeed a different way to construct persistence diagrams and our novelty is in the interpretation of ordinary persistence diagrams.  Other than our intuitive motivation in our previous response (2 in the list), unfortunately, we do not have a formal justification why we should treat points in a PD this way. Whether or not there exists some hyperbolic trend in the distance between persistence homology modules, as you commented in your previous response, is a very interesting question and a good direction for future search.
> > >
> > > We have added these two directions for future work in the conclusion.
> > >
> > > Regarding the modification on Fig.2, we had added a small label to indicate that the representation of the persistence diagram belongs on the Poincare ball (blue color, above the arrow). We newly added a sentence in the caption to emphasize that.
> > >
> > > Regarding your comment on a niche application, we have thoroughly thought of that and tried to identify a case where our method would be particularly good. We observed that the hyperbolic embedding outperforms all other baseline methods for the IMDB datasets, which are known to be smaller graphs compared to the REDDIT ones. Therefore, we hypothesize based on this that it could be better suited for smaller networks. We have added this at the end of Sec. 4.1.

---

### Official Review · AnonReviewer1 · 2020-10-30
**Insufficient Experiments**

**Rating:** 6
**Confidence:** 4

**Review:**

In this paper, the authors proposed a new representation of persistence diagrams that can include `''essential features''. Essential features correspond to the intrinsic topology of the underlying space that will not die during the filtration. To include the fact that the essential features are infinitely far from other normal features in the diagram, the authors proposed to use a Poincare ball representation, which maps the diagram into a disk whose boundary is infinitely far from inside.

The authors further proposed a classifier that learns the parameterization of the embedding of a diagram in the Poincare ball. The presentation learning procedure seems to be similar to (Hofer et al. 17), except for using a Poincare ball representation instead of the Euclidean square/triangle representation. Experiments are carried on graph classification tasks and image classification task.

On the positive side, I think the proposed representation well unified essential and non-essential topological structures. It is elegant and well-thought. A stability theorem is proven (in a similar manner as other known representations). The references are also reasonably complete.

However, I am having doubts on the practical motivation of the representation in the learning context. To me, essential features can be considered by simply adding the histogram of their birth times as additional features (to the neural networks at an appropriate layer). I think this approach is a natural and necessary baseline to be compared with.

Generally, the empirical results are not particularly strong. On graph datasets, the proposed method is only winning 2 out of 5 times. Many important baselines are also missing. For graph classification methods, state-of-the-art classifiers such as GIN and GraphSAGE should be at least compared with. For topological classifiers, many kernel methods could be compared with: PWGK, PI (both of which were used in the other experiment), also Sliced-Wasserstein Kernel. I honestly think an ensemble of these methods and the histogram of birth times of essential diagrams can be easy to tune and perform better.

Other questions/comments:

The references are not standard and need to be fixed.

For graph with Rips filtration, why are there 1-dim essential homology? Wouldn't all 1D homology features be killed eventually?

Details of the classifier (how the representation is learned) is still not clear even after reading the section in the supplemental material.

Experimental details are missing. I understand this is 80% training 20% testing. But how many folds were used to evaluate? Some baseline numbers are very similar to the numbers in the GIN paper. However, the GIN paper was using a 10-fold cross-validation. So there is some discrepancy in the experiments. I would appreciate if the authors could kindly elaborate.

Overall I think this is a nice mathematical formulation to incorporate essential features into the representation of persistent homology. But the practical usage in learning is not very convincing. Fundamentally, the essential features are completely different from non-essential ones. There is nothing in between them. Thus the benefit of a unified representation does not bring much more information than simply treating them separately.


** After rebuttal:
I am increasing the score to 6. I appreciate the authors' response to the reviews. They did the additional experiments I asked for.

As I stated in the original review, I really liked the unified approach. It is elegant and is nicely presented.

After reading the authors' response to R4, it is clarified that the 1D essential homology is because the computation over all threshold is too expensive. I think there might be an opportunity to better justify this paper: we often have to stop the filtration early due to computational concern. This unified representation could potentially be a good solution for this: without computing the actually death time, the unified solution can still 'learn' the real death time of the 1D essential classes. The authors might want to discuss or ideally empirically verify this in the final version. For example, can you show that using the new approach, and stopping earlier during the filtration, the unified classifier can be as good as when we run the whole filtration and compute the real death  time for all 1D classes. Moreover, it will be ideal if the authors can manage to show that the unified approach can actually learn the real death time for these fake essential classes (I do not know how). This way the paper can potentially have a bigger impact.

---

> ### Author Response · Authors · 2020-11-17
> **Response to Reviewer Feedback**
>
> We would like to thank the reviewer for this feedback and accept his criticism regarding the motivation and the lack of sufficient experiments to demonstrate our motivation. We have revised the paper following your suggestions and hope that the updated version addresses your concerns.
>
> We have added the baseline that you recommended. We created a paragraph in the experimental section and consider a histogram-based feature separation baseline. While including essential features via the histogram of their birth times is natural and simple to implement, unfortunately, it does not yield the best results. We agree that there may be a combination of methods (e.g., PWGK + histogram of birth times for essential features) that performs better than our method. An important benefit of our approach is, as you noticed, the unified representation. Therefore, we do not need to hand-tune “components” of other methods to obtain good performance. Also, we added an ablation study which demonstrates that the performance of our method is driven by the hyperbolic embedding.
>
> We have also added more state of the art results in the graph case study. Admittedly, our method does not achieve the best performance on all datasets. Our comparisons with other methods act as a proof-of-concept rather than an attempt to find the “universal best”. Please note that despite comparing several state of the art methods, it is hard to pick one that performs “best” on all benchmarks. We hope that you will agree that the facts that
>
> 1) our representation is unified.
> 2) the benefit of hyperbolic representation has been demonstrated via the ablation study,
> 3) the simplified feature separation baseline performs poorly irrespectively of the inclusion of essential features,
> 4) our method performs better or on par with several state of the art methods
>
> suffice to lend merit to our approach.
>
> The number of folds is 10 and we added that to the experimental section. The representation is learned using standard gradient methods. We highlighted that at the beginning of the experimental section and the full details are given at the end of the appendix.
>
> Regarding your comment on the essential features of the Rips filtration, please look at the follow-up response to Reviewer 4.

---

### Decision · Program_Chairs · 2021-01-07
**Final Decision**

**Decision:**

Accept (Poster)

**Comment:**

The paper proposes a novel method for representing persistence diagrams by embedding them to a Poincare ball.  The representation is learnable, and unifies essential and non-essential features.   The experimental comparisons with existing representation methods show significant improvements in performance.
The flexible data-driven embedding to a suitable geometric space is a novel idea, which will certainly advance the usefulness of TDA.  The  experimental resutls demonstrate well the advantage of the proposed repesentation.  The authors have also addressed the review comments appropriately, with some extra experiments.  This is also a good addition.